# Comparison of Measurements of External Load between Professional Soccer Players

**DOI:** 10.3390/healthcare10061116

**Published:** 2022-06-15

**Authors:** Roghayyeh Gholizadeh, Hadi Nobari, Lotfali Bolboli, Marefat Siahkouhian, João Paulo Brito

**Affiliations:** 1Department of Exercise Physiology, Faculty of Educational Sciences and Psychology, University of Mohaghegh Ardabili, Ardabil 56199-11367, Iran; r.gholizadeh60@yahoo.com (R.G.); l.bolboli@uma.ac.ir (L.B.); m_siahkohian@uma.ac.ir (M.S.); 2Department of Physical Education and Sport, Faculty of Sport Science, University of Granada, 18010 Granada, Spain; 3Department of Motor Performance, Faculty of Physical Education and Mountain Sports, Transilvania University of Braşov, 500068 Braşov, Romania; 4Faculty of Sport Sciences, University of Extremadura, 10003 Cáceres, Spain; 5Sepahan Football Club, Isfahan 81887-78473, Iran; 6Sports Science School of Rio Maior, Polytechnic Institute of Santarém, 2040-413 Rio Maior, Portugal; 7Life Quality Research Centre, 2040-413 Rio Maior, Portugal; 8Research Centre in Sport Sciences, Health Sciences and Human Development, 5001-801 Vila Real, Portugal

**Keywords:** external training load, technology, soccer, performance, wearable inertial measurement units

## Abstract

Background: The excessive and rapid increases in training load (TL) may be responsible for most non-contact injuries in soccer. This study’s aims were to describe, week(w)-by-week, the acute (AW), chronic (CW), acute:chronic workload ratio (wACWR), total distance (wTD), duration training (wDT), sprint total distance (wSTD), repeat sprint (wRS), and maximum speed (wMS) between starter and non-starter professional soccer players based on different periods (i.e., pre-, early-, mid-, and end-season) of a full-season (Persian Gulf Pro League, 2019–2020). Methods: Nineteen players were divided according to their starting status: starters (*n* = 10) or non-starters (*n* = 9). External workload was monitored for 43 weeks: pre- from w1–w4; early- from w5–w17; mid- from w18–w30, and end-season from w31–w43. Results: In starters, AW, CW, and wACWR were greater than non-starters (*p* < 0.05) throughout the periods of early- (CW, *p* ≤ 0.0001), mid- (AW, *p* = 0.008; CW, *p* ≤ 0.0001; wACWR, *p* = 0.043), or end-season (AW, *p* = 0.035; CW, *p* = 0.017; wACWR, *p* = 0.010). Starters had a greater wTD (*p* ≤ 0.0001), wSTD (*p* ≤ 0.0001 to 0.003), wDT (*p* ≤ 0.0001 to 0.023), wRS (*p* ≤ 0.0001 to 0.018), and wMS (*p* ≤ 0.0001) than non-starters during early-, mid-, and end-season. Conclusion: Starters experienced more CW and AW during the season than non-starters, which underlines the need to design tailored training programs accounting for the differences between playing status.

## 1. Introduction

Soccer is a sport with unstructured movement patterns, in which high-intensity performance-related energy systems are significantly involved [1]. During a competitive game, the total distances and sprints performed by professional soccer players are higher when compared to semi-professional soccer players. This reflects the different physical demands between soccer categories [2]. It is very important for coaches to pay attention to individual differences, such as physical fitness factors, workload responses, training intensity types, playing position, etc. [3,4], in order to improve the quality of teams’ activities and to reduce injuries [5]. Hence, the non-functional overreaching syndrome (NFOR) that could reduce team and player performance and increase club costs justifies the recent attention of researchers to avoid injuries [6,7,8,9]. Birrer reported (2013) that the NFOR/OTS career prevalence rate of Swiss elite athletes can be estimated at approximately 30%. In reviews of adult studies, using self-report questionnaires, prevalence rates for OTS/NFOR have been reported between 10–60%, with a predominance reported between 25–30% [10,11]. 

Soccer as a sport also requires variety of ballistic concentric and eccentric muscle contractions to perform a variety of movements, such as acceleration, deceleration and change of direction, tackling, jumping, and shooting, which increases the need for physiological and metabolic stress to support these movements. Soccer players accumulate fatigue from training and weekly matches during a competitive season, and when training/competition volume increases with inadequate recovery, players may enter NFOR mode, which is characterized by decreased performance [11]. Dietary and ergogenic studies have been reported as an intervention in athletes’ programs to achieve faster adaptation and manage exercise-induced fatigue. Sports coaches and researchers have the opportunity to reduce the risk of injury in athletes by manipulating some of these risk factors [8]. Training load (TL) is one modifiable risk factor that has recently received a lot of research attention. TL includes training intensity and duration, and research has shown that a higher TL increases the level of workload parameters, such as the acute:chronic workload ratio (ACWR), which leads to more non-contact injury occurrences and greater injury severity [12,13,14,15]. Some evidence has reported that many non-contact injuries occur when an athlete has a large amount of TL, in which case, any training or competition has the potential for athletic injury, indicating that an unfit workload can increase injury risk [16,17,18,19].

Lu et al. reported that professional soccer players had more total and relative exposure in the three weeks prior to injury [20]. However, an another perspective contends that TLs that underprepared players for the demands of match play could put them at risk of injury, and impede results [21]. Therefore, optimal training stimulus is crucial to maximize efficiency by using an acceptable TL while also limiting the detrimental effects of undertraining and overtraining [22]. TL can be considered as an internal or external load. The objectively assessed work performed by the athlete during training and/or competition, irrespective of internal workloads, is referred to as external TL [23]. Today, with the advancement of science, up-to-date and various tools are available to increase the efficiency of teams. Among these tools, we can see the reason for training monitoring. Devices such as GPS provide extensive information, such as acceleration and deceleration of speed, total distance in training, maximum heart rate, maximum velocity, etc., to coaches and team analysts [19]. Speed, acceleration, and distance covered are some of the most common external TL assessments. Internal TL represents an individual athlete’s response to training, and can be quantified by the intensity and duration of the physiological stress imposed on the athlete. Despite the advantages of both internal and external TL, it has been proposed that combining the two approaches could be the most efficient way to control TL [13]. Using this information, coaches can monitor player progress in training sessions, assess perceived training pressure levels on players, fatigue, and recovery, and use this information to design training sessions for future sessions. They can also use this information to identify the best players in each position who have the best physical condition during training, and place them in the starting line-up. The acute:chronic workload ratio (ACWR), a relative measure derived from Hulin et al., is a significant recent advancement in exploring TL [24]. Hulin et al. suggested the ACWR, which separates an athlete’s recent TL (i.e., acute workload: AW) by their TL over a long period of time (i.e., chronic workload: CW). This metric is suggested as a tool to help practitioners control TL within clear limits. The original idea was to hold ACWR within an arbitrary “optimal range” of 0.8–1.5 by preventing abrupt changes in TL. Understanding the workload–injury relationship is crucial to improve player availability and results. There is a contradiction of investigation on the association between workload and damage in professional soccer. The International Olympic Committee (2016) published a consensus statement that suggests that the use of the ACWR approach is effective in preventing non-contact injuries [25]. Nobari et al. have been reported that sprint variables (i.e., total distance, repeat sprint, or sprint total distance) may be related to the odds ratio of non-contact injuries in professional soccer players [14]. However, Impellizzeri et al. [26] concluded that there is no evidence that ACWR is used to manage TL and reduce the risk of injury. On the other hand, other researchers believe that excessive and rapid increases in TL may be responsible for most non-contact injuries [19,27]. Therefore, despite its increasing popularity as a load monitoring system, the ACWR and the injury risks associated with it need to be investigated further. Thus, the aims of this analysis were to: describe, week(w)-by-week, the AW, CW, and wACWR between starter and non-starter professional soccer players using the body load (BL); analyze differences in the AW, CW, and wACWR on a weekly basis between starter and non-starter professional soccer players based on different periods of the full-season (pre-, early-, mid-, and end-season); and analyze weekly total distance (wTD), weekly duration training (wDT), sprint total distance (wSTD), and repeat sprint (wRS) by season periods between starter versus non-starter professional soccer players. We hypothesized that starters would record greater workload and sprint variables during all periods of a soccer season compared to non-starters. 

## 2. Materials and Methods

### 2.1. Study Design

This study included a full season of a professional football team for 43 weeks in the Persian Gulf Premier League and knockout tournament during the 2019–2020 season. The 43 weeks of the full season were divided into 4 periods: pre-season (W1 to W4), early-season (W5 to W17), mid-season (W18 to W30), and end-season (W31 to W43). The three periods of the in-season have been considered as exactly the same number of weeks [28]. Forty-eight matches and 229 training sessions were held during the full season, which includes 11 congested weeks (i.e., two matches within 7 days) and 26 non-congested weeks (i.e., one match within 7 days). These periods were used to analyze the differences between starters and non-starters for AW, CW, wACWR, wTD, wSTD, wRS, wDT, and weekly maximum speed (wMS). Nineteen players were divided into two groups according to their starting status: starters (*n* = 10) and non-starters (*n* = 9). This criterion, based on a previous study, defined starter and non-starter players according to the time of participation in matches and training week-by-week [29]. Moreover, due to the outbreak of the COVID-19 pandemic, the team’s training between the 34th and 35th weeks was closed for four months from 28 February 2020 to 28 June 2020, and we have not considered this course due to the closure of training. On the other hand, due to the club’s decisions to maintain the level of physical fitness of the players during the lockdown from the COVID-19 pandemic, a training program video was provided for each player, and daily reports of training performance were sent by the players.

### 2.2. Participants

The present study, as a longitudinal descriptive study, was conducted throughout 2019–2020 in the Iranian Premier League. Nineteen professional soccer players of one club (28 ± 4.6 years; 181.6 ± 5.8 cm, 74.5 ± 5.6 kg) took part in the study during the full season, which was 43 weeks long. This research has been done by the club’s training coaches after coordination with the club managers and the head coach. The selection of players to participate in this study includes: continuous participation in competitions and training sessions during the season (when a player did not play in the weekly competition, they performed an alternative training session such as high-intensity interval training or a small side game); players could not be cross-trained during the study period; and players were not injured for more than two weeks. Based on previous studies [30,31], goalkeepers were not included in the report because of physiological variations in preparation and competition. The players participating in this study were divided into two groups, starter and non-starter. Starter and non-starter players were defined on a weekly basis. If a player suffered an injury and was unable to play and train weekly, he would be removed from the starting line-up in the same week. Before starting the study, players were introduced to the objectives of the study, and the consent forms were signed. The project was approved with the ethical code IR.ARUMS.REC.1399.546 by the Ardabil University of Medical Sciences, and is in compliance with the Declaration of Helsinki for human subjects [32].

### 2.3. External Training Load

#### 2.3.1. GPS 

Players’ daily physical activity was monitored using a GPS (GPS SPORTS systems Pty Ltd., Model: SPI High-Performance Unit (HPU), made in Canberra, Australia). This GPS model’s features include the following: 15 Hz position GPS, distance, and speed measurement; accelerometer: 100 Hz, 16 G Tri-Axial-Track impacts, accelerations, and decelerations, as well as data source body load (BL); Mag: 50 Hz, Tri-Axial; dimensions: smallest device on the market (74 mm × 42 mm × 16 mm); robustness; SPI high-performance unit based on Mining/Industrial Strength Electronics design; water resistance; and data transmission: infra-red and weighs 56 g [33]. Previous studies have shown that GPS can provide valid and reliable estimates of instantaneous and constant velocity movements during linear, multidirectional, and soccer-specific activities [24]. 

#### 2.3.2. Data Collection by GPS

The GPS information was used to measure external TL speed activities in all training and competition sessions. For every training session, all players used the same GPS to avoid mid-season takes between units. To allow access to satellite signals, and synchronize the GPS clock with the satellite atomic clock, all the devices were triggered 30 min before data collection. The information was downloaded to a computer after the recording, and analyzed using the software kit (GPS software, SPI High-Performance Unit (HPU), Canberra, Australia). GPS data were intended for the main team session (i.e., the beginning of the warm-up to the end of the last organized drill). In this study, baseline regions of SPI IQ absolute values were defined, and then, variables were assessed for each of the four time periods. 1. TD, 2. BL, 3 RS, 4.MS, 5. STD, and 6. BL reflects the quantity and intensity of acceleration events calculated from the accelerometer data. BL also acts as an integrated load variable used as an alternative TL marker (BL) for the original GPSports BL variable, and as a task speed marker (BL/min) as a TL criterion. 

#### 2.3.3. Calculate Training Load

BL is calculated from accelerometer data from GPSports devices taken from 3-axis accelerometers (planes X, Y, Z), and reflects both the amount and intensity of acceleration performed by the player. The following steps were performed in the BL calculation for each acceleration level: Initialize the BL counter to 0; Calculate the magnitude of the acceleration vector (V) of the current acceleration (V = ax2 + ay2 + az2); Normalize the magnitude vector (NV) by subtracting the country’s 1 G (NV = V 1.0 G). The unscaled BL (USBL) was then calculated using the formula USBLC = NV + (NV3) [19]. Next, the accelerometer recording speed (100 Hz) and load factor (EF) (SBLC = USBLC/100/EF) were used to calculate the scaled BL (SBLC) [34]. Finally, the final BL (BL = BL + SBLC) was determined. BL was replacing the original GPS port. Regular TL data were analyzed to report weekly load adjustments during the match season (AW, CW, and ACWR). The CW represents the rolling exponential of average accumulated training load of training session experience in the previous three weeks [35]. The AW represents the total of the training load experienced in the previous seven days [36,37]. ACWR represents the used uncoupled formula [38], which is defined in the equation below. All variables were calculated in each week of the experimental period, which is defined in the equation below [39].
(1)$$\mathrm{Uncoupled}\;\mathrm{ACWR}4=\frac{\mathrm{AW}4}{0.333\times \left(\mathrm{AW}1+\mathrm{AW}2+\mathrm{AW}3\right)}$$

ACWR was not computed for the pre-season period, and CW started from the third week according to the above formula. However, cumulative loads were used to calculate the ratio of CW and ACWR during the first few weeks of the season.

### 2.4. Statistical Analysis

Statistical analysis data were analyzed using IBM SPSS Statistics for Windows, Version 22.0 (IBM Corp., Armonk, NY, USA) statistical software package. The Shapiro–Wilk and Leven’s tests were used separately to evaluate the normality and homogeneity of variances to analyze data. The results were reported as mean ± standard deviation, with a 95% confidence interval (CI). Changes between the four in-season periods were assessed using a repeated-measures analysis of variance (ANOVA), followed by a Bonferroni with adjustment *post-hoc* test for pairwise comparisons. Partial eta-square (η2p) was calculated as the effect size of the repeated-measures ANOVA, and if the variable was not statistically normal, the Kruskal–Wallis H test was used to analyze the intergroup differences. Moreover, the effect size of Hedge’s g (95% CI) was reported. The Hopkins threshold was used to calculate the effect size [40] as follows: <0.2 = trivial, 0.2 to 0.6 = small, >0.6 to 1.2 = medium, >1.2 to 2.0 = large, >2.0 to 4.0 = very large, and >4.0, almost perfect. The significance level was considered at *p* ≤ 0.05.

## 3. Results

Figure 1 shows the changes in the AW variations over a full season with different periods (pre-, early-, mid-, and end-season) for starters and non-starters. The highest load in these variables was in w 35 (1303.9) arbitrary units (A.U. for starters). Besides, the highest values for non-starters in AW were in w 34 (1379.3 A.U.), whereas the lowest load was observed in w 29 (28.0 A.U.).

Figure 2 shows changes in the CW variations during the full season based on periods (pre-, early-, mid-, and end-season) and between weekly coefficient of variation (%CV) for this variable based on groups (starters and non-starters). The highest load in these variables was in w 38 (1073.7 A.U.); while the lowest load was observed in w 24 (509.0 A.U.) for starters. Besides, the highest values for non-starter CW were in w 5 (968.4 A.U.), whereas the lowest load was observed in w 29 (160.0 A.U.).

Figure 3 shows an overall vision of ACWR variations across the full season and its different periods (pre-season, early-season, mid-season, and end-season) for starter and non-starter players. Overall, the highest ACWR occurred in w 35 (2.3 A.U.) and w 21 (4.8 A.U.) for starters and non-starters, respectively; however, the lowest ACWR was found in w 11 and 33 (0.6 A.U.) for starters and w 30 (0.1 A.U.) for non-starters, respectively (Figure 3B). 

The results of the repeated-measures ANOVA revealed differences between season periods in AW starters (*p* < 0.001, η2p = 0.886), AW non-starters (*p* < 0.001, η2p = 0.944), CW starters (*p* < 0.004, η2p = 0.833) CW non-starters (*p* < 0.001, η2p = 0.957), ACWR starters (*p* < 0.003, η2p = 0.771), and ACWR non-starters (*p* < 0.04, η2p = 0.582). Theresults revealed significant greater weekly AW and CW and ACWR of starters compared to non-starters during the mid-season season (AW: *p* = 0.008, *g* = 2.36; CW: *p* ≤ 0.0001; *g* = 2.17; wACWR: *p* = 0.043, *g* = −0.95) and end-season (AW: *p* = 0.035, *g* = 0.91; CW: *p* = 0.017; *g* = 1.0; wACWR: *p* = 0.010, *g* = −0.92). Moreover, there was a significant difference in CW (*p* ≤ 0.0001, *g* = 2.18) between starters and non-starters in the early season. However, non-significant differences between groups were found for AW and CW during the pre-season, but not for AW and wACWR during the early season (Table 1).

Results of repeated-measures ANOVA revealed differences between season periods in starters wTD starters (*p* < 0.001, η2p = 0.966), wTD non-starters (*p* < 0.001, η2p = 0.982), wSTD starters (*p* < 0.003, η2p = 0.683), wSTD non-starters (*p* < 0.001, η2p = 0.953), wDT starters (*p* < 0.001, η2p = 0.965) and wDT non-starters (*p* < 0.001, η2p = 0.993), wMS starters (*p* < 0.001, η2p = 0.978) and wMS non-starters (*p* < 0.001, η2p = 0.992), and wRS starters (*p* < 0.02, η2p = 0.713) and wRS non-starters (*p* < 0.03, η2p = 0.749). The between-group comparisons for derived-GPS variables of distance and sprint in the different periods of the season are displayed in Table 2. Overall, the results showed that there was a significant difference between groups starters and non-starters for wTD (average value) (*p* ≤ 0.0001, *g* = 3.49 to 2.01), wSTD (*p* ≤ 0.0001 to 0.003; *g* = 2.88 to 1.55), wDT (average value) (*p* ≤ 0.0001 to 0.023; *g* = 3.34 to 1.09), wMS (average value) (*p* ≤ 0.0001; *g* = 3.48 to 2.07), and wRS (*p* ≤ 0.0001 to 0.018; *g* = 2.17 to 1.15) in the early-, mid-, and end-season. However, there were no differences between starters and non-starters for wTD, wSTD, wDT, wMS, wRS during the pre-season. 

## 4. Discussion 

The goals of this study were to: (i) identify, week-by-week, the AW, CW, and wACWR between starter and non-starter professionals soccer players using the BL; (ii) examine the aforementioned details related to the importance of quantifying the external TL over the season while attending to the starting status of the players; (iii) compare the gaps in AW, CW, and wACWR in BL, as well as the weekly average of distance and sprint variables, between starters and non-starters over the four season periods (pre-, early-, mid-, and end-season). On the other hand, in the present study, with the closure of the Persian Gulf League due to the COVID-19 epidemic, matches and group training were closed during this period, and training continued individually at home. Therefore, this time period was excluded from the present study.

The results of the BL analysis showed that both groups of players achieved the highest amount of AW and CW during the pre- and end-season. In non-starter players, the highest amount of wACWR was observed in the end-season, whereas in starter players, the highest amount was obtained in the mid-season. The fluctuations of AW and CW changes were consistent in the starter and non-starter players during the competitive season. However, the intergroup comparison showed that starters experienced more CW and AW during the season than the non-starters. Throughout the season, AW values were higher among non-starter players. Since the starter players are the starters of the matches, and for most of the playing time, these players are present in the match, it can be justified that these values are higher compared to non-starters, and also, this result could be related to the different levels of physical fitness of the players [41]. Other reasons include additional physiological stressors (high-speed running and number of sprints) [42] or psychological stains (rating of perceived exertion) [43] during a competitive season, which leads the starter group to increase its weekly training load and have a higher weekly workload than non-starters. On the other hand, perhaps more training can be attributed to the common strategy of the team’s technical staff to compensate, after the match or immediately after the match-play, for the participation of players who played for less than 45 min in an official game [30]. According to the contradictory results of recent research, the concept of ACWR, which indicates the possibility of injury, is questioned [26]. 

Changes in ACWR in the early-season were close to each other and there was no significant difference between groups, whereas in the mid- and end-season, there was a significant difference between groups. Some studies have shown an increased risk of injury with greater workloads [35,44] because exposure to a higher load is inevitable. Besides, Bowen et al. pointed out that an acute, excessive, and rapid increase in loads compared to a higher chronic load may be responsible for a large proportion of non-contact injuries [45]. Thus, non-starters should be prepared to experience greater stress during periods when they are not participating as starters in matches in order to reduce their injury risk [46]. Curtis et al. demonstrated that non-starters had higher training workloads, as measured from total covering distance during weekly training sessions [47]. Researchers attribute the workload difference between starters and non-starters to a variety of factors, such as age, competitive environment, tactic strategy of the league, length of the season, and league fixture [30]. However, in another part of the results, in non-starters, ACWR in the mid- or early-season were at 1.4 and 1.6, respectively. Bowne et al. reported a significant increase in injury risk (relative risk = 2.6) when CW was low (<938 m) and when ACWR was high at high-speeds (1.4–1.9 A.U.). These results, though seemingly not generalized, suggest that ACWR surveillance can be an important strategy to prevent injury in professional soccer [17]. Most studies on the relationship between workload and injuries have removed contact injuries, as they appear to be unavoidable, but many researchers have suggested that ACWR is a good variable of the risk of non-contact injury in players [45]. Recent research has questioned the concept of ACWR as a marker to identify possible injuries [48,49]. No one criticizes the true value of understanding changes in TL progression or changes over the course of a few weeks. Interestingly, training heterogeneity can lead to ACWRs that differ from play position differences during the season, as they are stagnant or show sensitivity for regular periods. Of course, the difference between these changes may depend on the type of measurement considered.

The results of the present study revealed that starters registered greater wTD, wSTD, wDT, wRS, and wMS during the early-, mid-, and end-season in comparison with non-starters in every season period. Anderson et al. and Oliveria et al. did not report the TD difference between starters and non-starters; however, the authors concluded that the high-intensity season loading pattern depended on the starting position of the players in favour of the starters in the English Premier League [46,50]. In line with previous studies, our results showed greater wDT and wTD in starters compared to non-starters during the season. Consistent with our results, the highest TD values were reported by researchers during the pre- and early-season compared to the mid- and end-season. Oliveria et al. attributed the great TD in the pre-season to the coaches’ emphasis on physical conditioning, which reduces the number of other courses compared to the pre-season. In addition to examining the external TL, some articles have also analyzed the internal TL over a season [50]. Researchers related the difference between starter and non-starter players to the starting position of the players, coach’s decisions for designing drills, play styles, and competition demands across leagues, which should be considered by the technical staff so that a suitable training strategy can be considered for each player [19,51]. It seems that the differences in GPS variables between starters and non-starters are understandable because only 11 players participate in each competition, so the other players are not exposed the load of the match, and so, these reported differences are justified. Managing starter and non-starter players is very important for the team manager and staff of the professional teams present during the tournament schedule. External load monitoring provides good reference information about the physical impact and physiological stresses during an annual cycle. It seems that awareness of training intensity, recovery status, and substitution strategies in systematic planning in a competitive season to prevent injury, overtraining syndrome, and NFOR is too important.

### Limitations of Study

Regarding the limitations of the present study, we can mention the lack of internal TL criteria. This study examines a team in the Persian Gulf League that has faced a shutdown due to the expansion of COVID-19. Due to the fact that we only had access to one of the 16 teams in the Premier League, so the number of samples was small, we could not report the sample size calculation [32]. A lack of physiological analysis in the training load portion was one of the limitations of the present study. Another limitation of this study is the fact that players trained at home during the COVID-19 league closure. Simultaneous examination of internal and external load may have been more appropriate. It is also possible in future studies to compare AW and CW orientation strategies for TL management in higher statistical samples.

## 5. Conclusions

According to the results, starters’ AW, CW, and ACWR were greater than non-starters. Moreover, the ACWR value of the starters in the mid- and end-season showed high numbers that should be considered by the technical staff to review the TL strategies. In the next section, the results of starters’ wTD, wSTD, wDT, wRS, and wMS variables in comparison to non-starters showed a higher mean, and also in both groups, compared to the pre-season, the average of all variables was reduced.

According to the results of this study, it seems that training monitoring using the parameters measured in this study can be a good way to check the external loads during training and competition. Moreover, coaches and professional soccer players can use the information obtained in this way to check the progress during the training season, review the performance during the match, and design appropriate training sessions.

## Figures and Tables

**Figure 1 healthcare-10-01116-f001:**
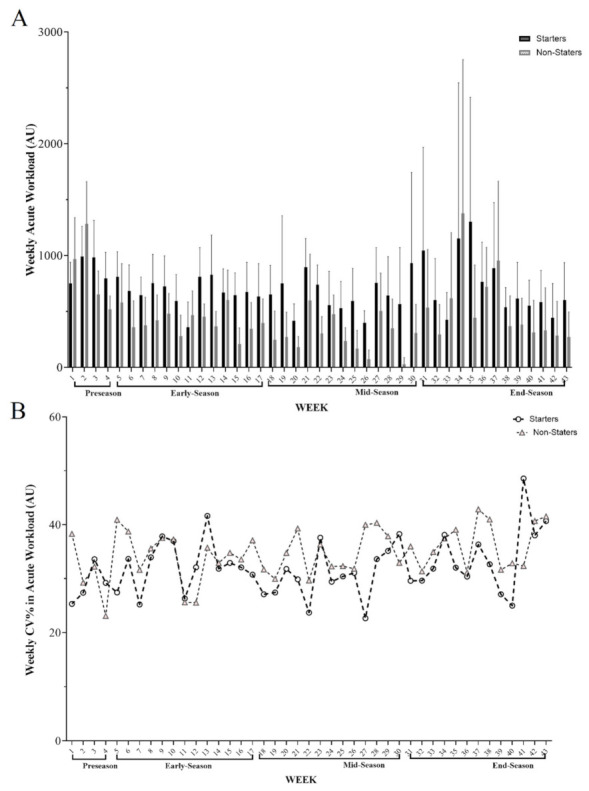
(**A**) Descriptive variations across AW (weekly acute workload) during the full season based on periods and (**B**) between weekly coefficient of variation (%CV) for this variable based on groups.

**Figure 2 healthcare-10-01116-f002:**
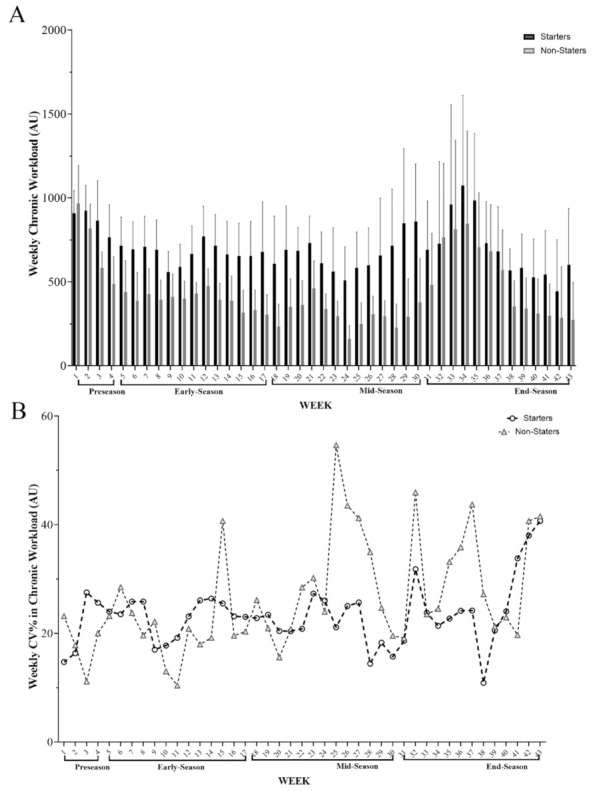
(**A**) Descriptive variations across CW (weekly chronic workload) during the full season based on periods and (**B**) between weekly coefficient of variation (%CV) for this variable based on groups.

**Figure 3 healthcare-10-01116-f003:**
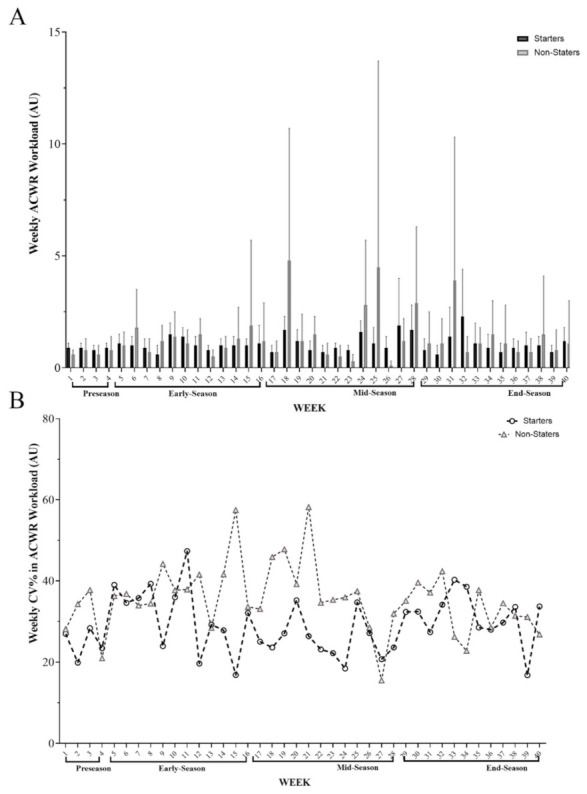
(**A**) Descriptive variations across wACWR (weekly acute/chronic workload) during the full season based on periods and (**B**) between weekly coefficient of variation (%CV) for this variable based on groups.

**Table 1 healthcare-10-01116-t001:** Differences between starters and non-starters for AW, CW, and wACWR related to body load in the different periods of the season.

Variables	Season Period	Group		% Difference (Non-Starters vs. Starters)	*p*	Hedges’s *g* (95% CI) (Non-Starters vs. Starters)	Inference
		Starters	Non-Starters				
**AW (A.U.)**	*Pre-Season*	880.9 (129.8)	856.0 (183.6)	2.4 (−12.7 to 7.7)	0.307	0.1 (−0.7 to 1.0)	small
	*Early-Season*	680.6 (151.9)	411.4 (56.1)	2.6 (1.5 to 3.8)	0.227	2.1 (1.0 to 3.3)	very large
	*Mid-Season*	649.8 (190.9)	288.5 (66.2)	3.6 (2.2 to 5.0)	0.008 *	2.3 (1.1 to 3.5)	very large
	*End-Season*	732.5 (258.6)	531.1 (137.9)	4.0 (−275.5 to 20.1)	0.035 *	0.9 (−0.0 to 1.8)	moderate to large
**CW (A.U.)**	*Pre-Season*	916.4 (132.6)	893.5 (182.8)	2.2 (−1.3 to 1.7)	0.756	0.1 (−0.7 to 1.0)	small
	*Early-Season*	696.8 (155.6)	425.4 (51.8)	2.7 (1.5 to 3.8)	<0.001 *	2.1 (1.0 to 3.3)	very large
	*Mid-Season*	637.3 (194.2)	301.3 (60.4)	3.3 (1.9 to 4.7)	<0.001 *	2.1 (1.0 to 3.3)	very large
	*End-Season*	752.7 (250.5)	526.3 (148.6)	2.2 (2.3 to 4.2)	0.017 *	1.0 (0.0 to 1.9)	large to very large
**wACWR (A.U.)**	*Early-Season*	1.0 (0.0)	1.0 (0.1)	−0.5 (−12 to 13)	0.133	0.0 (−0.9 to 0.9)	trivial
	*Mid-Season*	1.1 (0.2)	1.6 (0.7)	−5.0 (−9.8 to −1.4)	0.043 *	−0.9 (−1.9 to −0.0)	trivial
	*End-Season*	1.10 (0.16)	1.4 (0.38)	−3.0 (−6.0 to 0.1)	0.010 *	−0.9 (−1.8 to 0.0)	trivial

Abbreviations: AU, arbitrary units; AW, weekly acute workload in AU; CW, weekly chronic workload in AU; wACWR, weekly acute:chronic workload ratio in AU; *p*, *p*-value at alpha level 0.05; Hedges’s *g* (95% CI), Hedges’s *g* effect size magnitude with 95% confidence interval. * Significant differences for *p* ≤ 0.05. Hedge’s *g* was interpreted as: <0.2 = trivial, 0.2 to 0.6 = small, >0.6 to 1.2 = medium, >1.2 to 2.0 = large, >2.0 to 4.0 = very large, and > 4.0, almost perfect.

**Table 2 healthcare-10-01116-t002:** Differences between starters and non-starters for derived GPS variables of distance and sprint in the different periods of the season.

Variables	Season Period	Group		%Difference (Non-Starters vs. Starters)	*p*	Hedges’s *g* (95% CI) (Non-Starters vs. Starters)	Inference
		Starters	Non-Starters				
**wTD** **(m)**	*Pre-Season*	4995.8 (399.9)	4755.8 (356.3)	240 (−1.2 to 6)	0.187	0.60 (−0.31 to 1.52)	trivial
	*Early-Season*	4745.9 (498.9)	2974.2 (466.7)	17.7 (1.3 to 2.2)	<0.001 *	3.49 (2.06 to 4.92)	very large to nearly perfect
	*Mid-Season*	4568.3 (593.2)	2385.7 (559.2)	218.2 (1.6 to 2.7)	<0.001 *	3.61 (2.15 to 5.06)	very large to nearly perfect
	*End-Season*	4116.3 (668)	2696.7 (679.9)	141.9 (7.6 to 2)	<0.001 *	2.01 (0.90 to 3.11)	very large
**wSTD** **(m)**	*Pre-Season*	2541.4 (479.7)	2546.6 (571.3)	−520 (−5.1 to 5.0)	0.883	−0.00 (−0.91 to 0.89)	trivial
	*Early-Season*	1869.3 (282.8)	1123.7 (199.4)	7.4 (5.0 to 9.8)	<0.000 *	2.88 (1.59 to 4.16)	very large
	*Mid-Season*	1782.5 (402.8)	854.8 (220.4)	9.2 (6.0 to 1.2)	<0.000 *	2.68 (1.44 to 3.92)	very large
	*End-Season*	1876.4 (425.9)	1194.7 (413.0)	6.8 (2.7 to 1.0)	<0.003 *	1.55 (0.52 to 2.57)	large
**wDT** **(min)**	*Pre-Season*	76.8 (5.8)	76.1 (4)	70 (−4.1 to 5.5)	0.704	0.13 (−0.76 to 1.04)	trivial
	*Early-Season*	67.2 (5.4)	48.9 (7.4)	1.8 (1.2 to 2.4)	<0.001 *	3.34 (1.94 to 4.73)	very large
	*Mid-Season*	59 (7.5)	37.7 (6.7)	2.1 (1.4 to 2.8)	<0.001 *	4.25 (2.63 to 5.88)	nearly perfect
	*End-Season*	51.8 (10.3)	39.3 (11.5)	1.2 (1.9 to 2.3)	<0.017 *	1.09 (0.127 to 2.058)	small to moderate
**wMS** **(km·h^−1^)**	*Pre-Season*	26.3 (2.5)	23.9 (2.3)	240 (0 to 4.7)	0.052	0.91 (−0.03 to 1.85)	Moderate to large
	*Early-Season*	28.5 (2.4)	18.8 (2.4)	97 (7.3 to 12)	<0.001 *	3.48 (2.05 to 4.91)	very large
	*Mid-Season*	26.1 (3.3)	17.1 (1.7)	90 (6.4 to 11.5)	<0.001 *	3.34 (1.954 to 4.74)	very large
	*End-Season*	23.9 (3.2)	18.5 (2)	54 (2.7 to 8)	<0.001 *	2.07 (0.95 to 3.18)	very large
**wRS** **(number)**	*Pre-Season*	88.3 (23.6)	75.4 (26.1)	1.2 (−1.1 to 3.7)	0.271	0.49 (−0.41 to 1.41)	small
	*Early-Season*	62.5 (9.5)	43.2 (7.2)	1.9 (1.1.0 to 2.7)	<0.001 *	2.17 (1.03 to 3.30)	very large
	*Mid-Season*	59.5 (10.0)	36.5 (8.9)	2.3 (1.3 to 3.2)	<0.001 *	2.31 (1.15 to 3.47)	very large
	*End-Season*	61.3 (13.3)	44.1 (15.3)	1.7 (0.3 to 3.1)	<0.016 *	1.15 (0.17 to 2.12)	large

Abbreviations: wTD, weekly total distance in meters; wSTD, weekly sprint total distance in meters; wDT, weekly duration training in minutes; wMS, the average accumulated of the maximum sprint is calculated and reported each week in kilometres per hour; wRS, weekly repeat sprint in meters; *p*, *p*-value at alpha level 0.05; Hedges’s *g* (95% CI), Hedges’s *g* effect size magnitude with 95% confidence interval. * Significant differences for *p* ≤ 0.05. Hedge’s *g* was interpreted as: <0.2 = trivial, 0.2 to 0.6 = small, >0.6 to 1.2 = medium, >1.2 to 2.0 = large, >2.0 to 4.0 = very large, and > 4.0, almost perfect.

## Data Availability

The datasets used and/or analyzed during the current study are available from the corresponding author on reasonable request.

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
