# Peer review of "Comparison of Measurements of External Load between Professional Soccer Players"

_healthcare, 2022, doi:10.3390/healthcare10061116_

Round 1

Reviewer 1 Report

Dear Authors,

Thank you for this well-designed study. With minor improvements, I believe that the manuscript can be a valuable contribution to the efforts of understanding the role of wearables to measure and compare the performance of soccer players.

You can find my recommendations below:

- Please give more details about why you need to perform this study and why the results of your study is relevant and important for the field.

- At the conclusion section, please provide point to point recommendations to players, trainers and sport scientists about the potential applications of your study into the training protocols and loading regimes that aim to improve athletic performance of soccer players.

Best wishes and stay healthy.

Author Response

Reviewer 1

Dear Reviewer
Thanks again for your comments and please see the attachment.
BR

Reviewer 2 Report

The authors described the weekly and seasonal differences in factors associated with training load between starter and non-starter professional soccer players using a wearable GPS device. The authors hope to use those findings to guide the development of tailored training programs for players with different playing statuses. 

The research topic is appropriate for the journal's scope. The longitudinal examination of the differences in the training load factors between different players could provide valuable information for training to promote performance and prevent injuries. The paper was well written and structured.

My only concern is about the statistical analysis. Since there are two groups with multiple time points of measurement, should the analysis of variances be used instead of t-tests? I noted that the authors are only interested in group differences, but running multiple t-tests would lead to the inflation of the type I error. In addition, are those outcome measures (AW, CW, wACWR; and wTD, wSTD, wDT, wMS, wRS) independent from each other? If not, then the multivariate analysis of variances should be used.

Minor issues: In table 1, there is an extra line under the pre-season data of AW.

Author Response

Dear Reviewer
Thanks again for your comments and please see the attachment.
BR

Reviewer 3 Report

1. The title may need to be shortened.

2. The background and methodology of the abstract need to be specified.

3. Rows 35-45: What is the prevalence of functional hyperextension syndrome between professional and semi-professional soccer players?

4. Lines 46-61: How to define a large number of TL

5. Lines 125-143: How to ensure cross-training during the study period, and players who have not been injured for more than two weeks.

6. Very large values in ES Inference, such as AW(AU) Early-Season, are not explained clearly in the discussion.

7. Goalkeepers should not be a research limitation for this article.

8. The lack of physiological analysis in the training load portion of this study may be a limitation of this paper.

9. Solo training at home This may be a limitation of this article.

Author Response

(The authors gave the same response as above.)
